# Impact of Seasonal Changes of Precipitation and Air Temperature on Clay Excavation

**Denis Težak [1],\* , Božo Soldo [2], Bojan Đurin [2] and Nikola Kranjčić [3]**

[1] Independent researcher, Jalkovečka 104A, 42000 Varaždin, Croatia
[2] Department of Civil Engineering, University North, Jurja Križanića 31b, 42000 Varaždin, Croatia; bozo.soldo@unin.hr (B.S.); bojan.durin@unin.hr (B.Đ.)
[3] Faculty of Geotechnical Engineering, University of Zagreb, Hallerova aleja 7, 42000 Varaždin, Croatia; nkranjcic@gfv.hr
\* Correspondence: dtezak@gmail.com

**Abstract:** Excavation of clay soil is one of the most important economic branches in the northern part of Croatia. The impact of clay soil in Croatia compared to the global exploitation fields of clay soil is negligible. Modern methods of clay excavation during winter months due to negligible amounts are not profitable. Therefore, it is important to optimize clay soil excavation throughout the year to increase the efficiency of exploitation and increase profits. In the case of large amounts of precipitation (rain), clay absorbs water and becomes grain. For this reason, access to the exploitation field and excavation itself becomes impossible. Air temperature also plays an important role in excavation. Long-lasting low air temperatures below 0 °C during the winter months result in clay frost. As a result, excavation cannot occur at that time. The paper describes a new method of modeling the precipitation and air temperature on the exploitation fields of clay in Northwest Croatia on the exploitation fields of Cukavec and Cukavec II. The method involves the calculation of the drought index and use of the rescaled adjusted partial sums (RAPS) statistical method and its application on a time series of total daily precipitation and average daily temperatures as a climatic indicator of any observed area. Using this process, it is possible to determine the time period of the year when clay soil can be excavated.

**Keywords:** exploitation clay fields; excavation; drought index; RAPS; precipitation; air temperature

## 1. Introduction

The excavation of clay soil for use as a construction material is an important economic resource in the northern part of Croatia, as well as many other regions in the world. The exploitation fields of clay in Croatia are small in size, ranging from 10 to 30 ha. Clay soil excavation requires the use of bulldozers, crawler-mounted hydraulic excavators, and dump trucks to excavate, stockpile, and transport material. Since excavation is based on relatively light excavators performing the work, conditions must be optimal for them to operate economically. Clay minerals can adsorb large volumes of water within their crystal lattice, often making them difficult or impossible to excavate and handle [1,2].

Due to the material limitations on excavations in late autumn, the winter and early spring periods may often become problematic. Possible remedies in the winter months, such as solar methods, steam jets, water needles, electric needles, valveless pulsejet engines or hydraulic methods [3,4], are only worthwhile for a large exploitation field, due to high initial capital costs. In addition to poor material conditions, other reasons for a possible delay or inability to excavate clay soil may be of a technical nature, i.e., unavailable or inoperative excavator, operator or a truck driver absence, additional works under a construction contract, and/or unpredicted orders, as well as cancellation or increased order.

Because of these issues, there is a need for a methodology that will help to determine appropriate time periods when excavation should proceed. Precipitation and air temperature are dominant meteorological indicators that determine the condition of the clay material and, hence, the likelihood of a timely and economical excavation sequence.

The territory of the Republic of Croatia is divided into three large natural and geographical areas: the Pannonian and Peripanian area, the hilly and mountainous area, and the Adriatic. The exploitation fields of clay soil are located in the Pannonian and Peripanian area, which includes lowland and hilly parts of Eastern and Northwest Croatia. In this area, precipitation is higher during the warm part of the year—the continental maximum rainfall in the summer period (April–September). This generally occurs as convection rain, often coupled with storms and winds [5,6].

The average air temperatures, the amount of rainfall, and their annual regime are determined by the climate regions of Croatia. They also represent the main climatic and ecological areas. According to generally accepted world climate categories, most of the area in Croatia includes a moderately hot rainy climate (type C), while only high mountain regions have a snow-forest climate (type D) [7,8].

Taking into account the climate of Northern Croatia, the ideal conditions for exploitation of clay soil are between mid-April and mid-September. From a technological point of view, precipitation and air temperature are constraining factors in planning and implementation of clay excavation. Precipitation impact in terms of infiltration, saturation of clay pores, and stability of clay and clay moisture has been analyzed in [9,10], while the impact of precipitation on the chemical composition of clay soil has been analyzed in [11]. Similarly, the influence of air temperature on clay soil excavation and the influence of increased pressure on air and stresses due to temperature increase have been analyzed in [12]. However, where the influence of precipitation and air temperature has been analyzed, the mechanical properties of clay soil are described without any regard for excavation technology or predicting a favorable period for clay excavation.

This is the motivation for developing a model that will define periods when conditions for clay excavation are favorable. For this purpose, calculation of a drought index ($J$) and $RAPS_k$ values, i.e., application of a rescaled adjusted partial sums (RAPS) method, will be applied to the time series of annual total precipitation and average daily air temperature. The drought index ($J$) is planned to be used for the purpose of identifying and monitoring drought periods during the analyzed years and months, with regard to the total precipitation and average daily temperatures. This defines the climate of the observed area [5–8], as well as the possibility for clay excavation. Rescaled adjusted partial sums would be used for determining possible subseries of the original time series of total daily precipitation and average daily temperatures. These subseries may help to identify characteristic periods in a year. It is expected that the combination of a drought index ($J$) and the RAPS method will define optimum periods for clay excavation. Accent should be on the ending of the period of the clay excavation, due to the finalization of the business plans of the excavation. The abovementioned methods have not been applied to the described problem so far and are presented and described below.

## 2. Methods

### 2.1. Drought Index (J)

There are many indicators used for defining and characterizing of the drought periods in a year. All of them can be applied, depending on climate characteristics and available data, as well as on the desired information (results). The standardized precipitation index (SPI) is the most commonly used indicator for defining a drought period [13]. SPI is very simple to calculate, because precipitation data are the only input values [14]. Despite this advantage, a drawback of SPI is that air temperature is not considered. The value of $P_{ed}$, [15] is also used for determining the frequency and severity of drought in Russia. Equation for $P_{ed}$ includes data about precipitation, air temperature, and potential evaporation. In general, such an indicator can be applied to any location, but a big problem is that evaporation is sometimes very hard to measure at particular locations. The effective drought index

(*EDI*) is used for examining the duration and strength of drought, by taking into the account only precipitation data [13]. Despite this, calculation of the EDI is not simple and requires definition and usage of many mathematical functions, which increases the complexity of the procedure. The standard index of annual precipitation (SIAP) is used for definition of drought intensity also by taking into account only precipitation. The calculation procedure is simple, but the methodology has been applied only to Iran [16]. There are many other indicators which can be used for description and defining the drought period during the analyzed time period. It is obvious that application of a particular indicator depends on the available input data, as well as the climate conditions of the observed area.

For the purpose of identifying and monitoring drought periods for the case study presented in this paper, the drought index (*J*) [17] was applied. The value of *J* was calculated using the average air temperature and total precipitation (Equations (1) and (2)) during the period of the clay excavation.

The drought index calculated for the period of one year is defined by:

$$J_y = \sum_{i=1}^{12} \frac{P_i}{T_i + 10} \tag{1}$$

where $P_i$ is the total monthly rainfall and $T_i$ is the monthly average temperature.

Further, the drought index can be calculated for a period of one month:

$$J_m = \frac{P}{T + 10} \tag{2}$$

Table 1 shows drought intensity ranges with respect to the drought index (*J*) values [17].

**Table 1.** Drought intensity related to the drought index (*J*).

| Degree of Drought | Drought Index (*J*) Values |
|---|---|
| Normal | ≥30 |
| Light drought | 20–30 |
| Moderate drought | 5–20 |
| Extreme drought | ≤5 |

## 2.2. Rescaled Adjusted Partial Sums (RAPS) Method

The rescaled adjusted partial sums method (RAPS) is based on time series analysis using the deviation sum curve. Visualization of the RAPS method is helpful since it smoothens small system and random errors in the time series. A graphic presentation of the RAPS method often reveals the existence of subseries with similar characteristics, larger scale trends, sudden value changes, irregular fluctuations, existence of periodicity within the time series analyzed, etc. The RAPS method is defined by the following expression:

$$RAPS_k = \sum_{t=1}^{k} \frac{Y_t - \overline{Y}}{S_y} \tag{3}$$

where $\overline{Y}$ is the average value of the entire time series, $S_y$ is standard deviation of the same series, *n* is number of data in time series, and *k* is summation counter ($k = 1, 2, 3, \ldots, n$) [18].

Such a transformation, i.e., graphical presentation of the $RAPS_k$ values will often point to the existence of regularities in the fluctuations of analyzed parameters ($Y_t$) [19]. The process of determining a new subseries is based on the visual determination of the highest "peak", or lowest "valley". When the existence of the subseries within the main series has been determined, the next step is to determine (as a general rule, used in a large number of analyses of time series of different data) the linear trends of the subseries, which is a usual procedure within the RAPS method.

The RAPS method has been used in many research areas, but hydrology is its most common application. Bonacci used this method in [20] for an analysis of the causes for different spring discharge

characteristics during this period. Flows and water temperatures of the River Danube at Bratislava in Slovakia were also analyzed in [21], as well as flows of River Lika and Gacka in carst areas of Croatia [22] and springs in carst areas of Italy [19]. There are many examples, i.e., Reference [23] for the flows and precipitations of the river basin of the river Sacramento in California, as well as Reference [24] for the analysis of the flows, deposits, and precipitation in river basins of the Chinese Rivers Weihe and Jinghe.

In Lojen S. et al. [25], RAPS was used for the analysis of the geochemistry characteristics of the river Krka in Croatia. In Đurin B. et al. [26] and Tadić L. [27], RAPS was applied to the time series of insolation, air temperatures, and precipitation for the purpose of irrigation. Waste water quality analysis can also be made by RAPS [28]. A wide application of the RAPS method can be seen from the literature review.

## 3. Methodology

Since the clay material in the exploitation fields of Cukavec and Cukavec II has been continuously exploited from April to October for more than fifty years, it is necessary to optimize the period for the excavation of clay material during the year. Clay is hydroalumosilicate; it is necessary to determine the favorable months of the year when exploitation of clay soil can be undisturbed. Given the stochastic aspects of the precipitation and air temperature, it is necessary to quantify and to predict precisely influence of the precipitation on clay excavation.

The first step involves calculation of the drought index $J_y$, (Equation (1)) for the period of one year, for each of 10 observed years, as well as for the average of 10 years. Averaging a ten-year set of total daily precipitation and average daily air temperatures ensures continuity and the required length of meteorological indicators that in fact define the period within which clay soil is excavated. The purpose of this step is to define variability at scale of characteristic years(s), as well as evaluation for possible analysis at smaller time periods. This presents motivation for calculation of the drought index $J_m$, (Equation (2)) for the period of one particular month for all observed years.

Once the initial RAPS method is applied, we can examine the results and identify any subsequences that may deserve further attention. If warranted, the subsequences may be further analyzed and compared to determine if there are periodic recursions of sequences.

## 4. Case Study

### 4.1. Location

The excavation of the clay soil is carried out at the exploitation field of Cukavec and Cukavec II. The owner is Leier-Leitel d.o.o., a limited liability company. The exploitation fields are located in Northwest Croatia near Varaždin (Figure 1). Table 2 shows the coordinates of the Cukavec and Cukavec II. To the north side of the Cukavec II is Cukavec, with the Cukavec stream running along its western side. There are no structures (buildings) in the exploitation field which might interfere with excavation.

The amounts of the clay reserves confirmed within Cukavec exploitation field on 31 December 2000 amounted to 14,621 m$^3$ which, in view of the planned batch production in the future, does not meet the demand for quality clay. Therefore, the exploitation continued on Cukavec II. The confirmed exploitation reserves for the Cukavec II exploitation field amount to about 670,000 m$^3$. According to the contract to excavate clay from Cukavec II from May 2018 [31], the annual minimal and maximal clay quantities to be exploited were 18,000 m$^3$ and 50,000 m$^3$, respectively. Since exploitation reserves are small, an extension of the exploitation field has been requested.

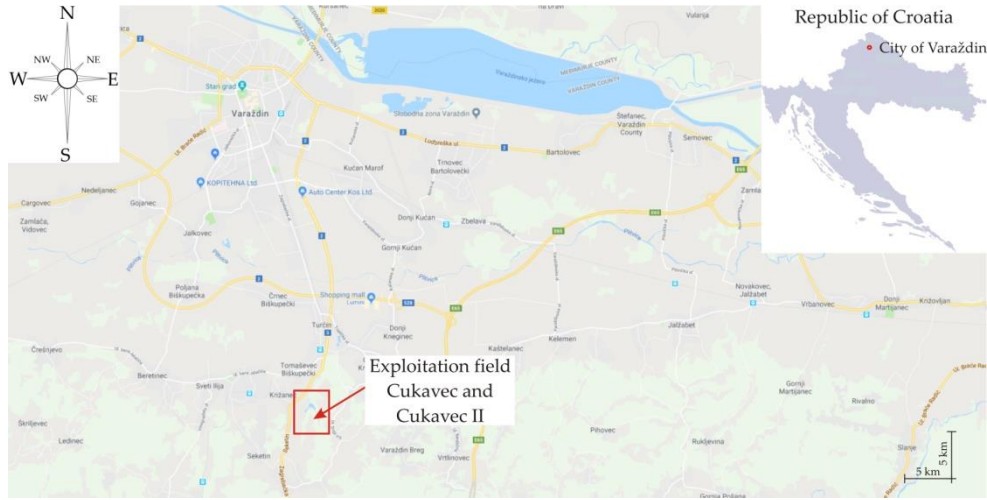

**Figure 1.** Location of the clay exploitation field Cukavec and Cukavec II [29,30].

**Table 2.** Coordinates of the exploitation field of Cukavec and Cukavec II (projection WGS84).

| $\varphi$ (d° m′ s″) | $\lambda$ (d° m′ s″) |
|---|---|
| 46° 14′ 32.09 v″ | 16° 21′ 31.87″ |

WGS84-World Geodetic System (The National Geospatial-Intelligence Agency (NGA), Springfield, VA, USA); $\varphi$-latitude; $\lambda$-longitude.

The clay is excavated on the Cukavec II surface, located directly next to the production facility of the building material (Figure 2). The clay is then transported to a processing plant, where it is cut and processed. The processed clay is further transported to landfill sites in a way that the layers of different thicknesses are matched depending on the platform being used, i.e., the quality and the physical properties of clay, thus creating a mixture of optimal quality for the further production process.

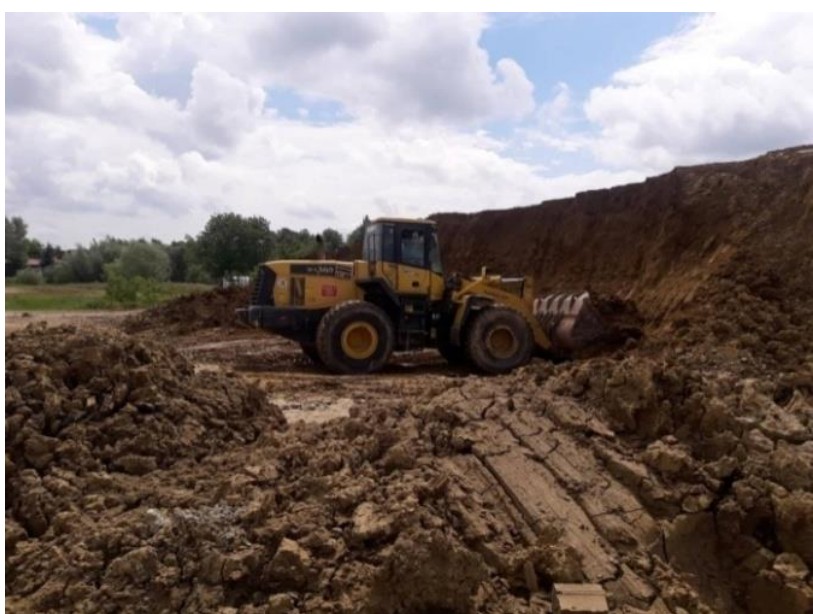

**Figure 2.** Exploitation field Cukavec II [31].

### 4.2. Input Data

Figure 3 shows the average air temperature over ten years from 2008 to 2017, while Figure 4 shows the average precipitation levels in 10 years, 2008–2017 [32,33].

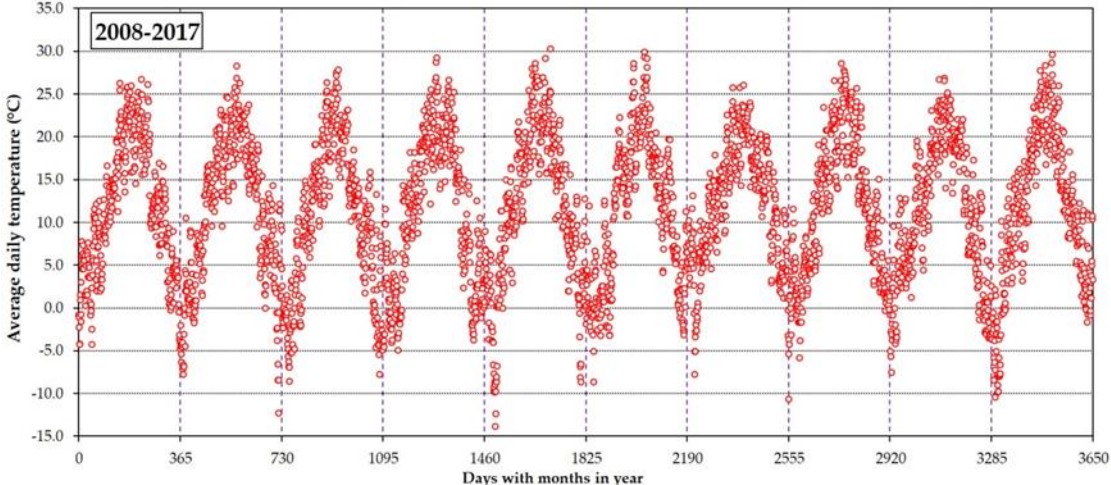

**Figure 3.** Average daily air temperature in the period from 2008 to 2017.

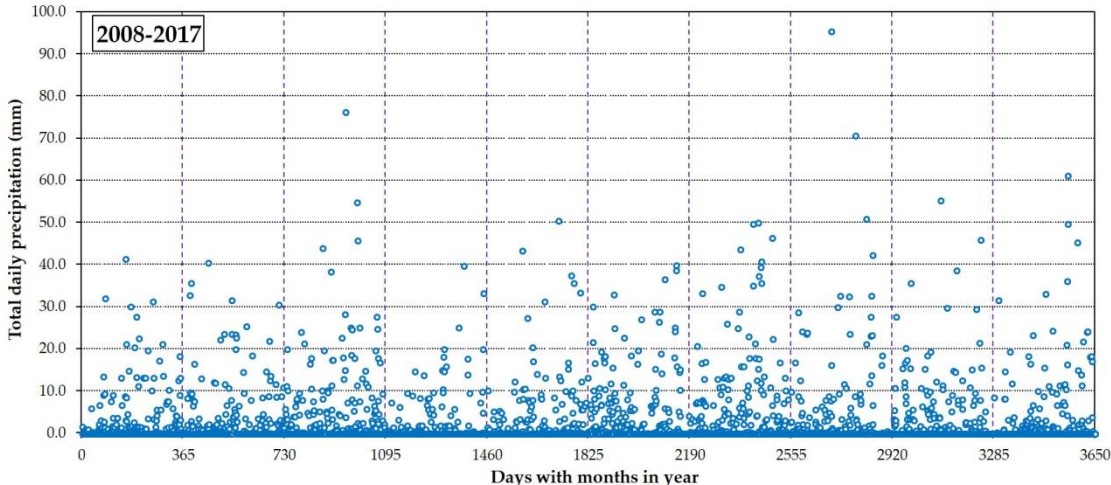

**Figure 4.** Total daily precipitation in the period from 2008 to 2017.

The average daily air temperature values are between the lowest (−13.8 °C, measured in February 2011) and the highest (30.3 °C, measured in July 2011). The values of the total daily precipitation amount range from 0 to the highest measured volume of 95.5 mm, recorded in April 2015. In this case, the definition of the trends is unacceptable due to the apparent data dispersion.

Both images show the stochastic nature, respectively, of a large dispersion of analyzed meteorological indicators, where defining the trend is not justified. In doing so, defining the functional entity of changing the average daily air temperature and total daily precipitation is not possible. Figures 5 and 6 show average annual values of temperature and precipitation. Compared to the averaged air temperatures, the jumps in the precipitation values are even more pronounced. Further steps would be calculation of the drought indexes, as well as the analysis of time subsets using the RAPS method, carried out by the Microsoft Excel spreadsheet calculator.

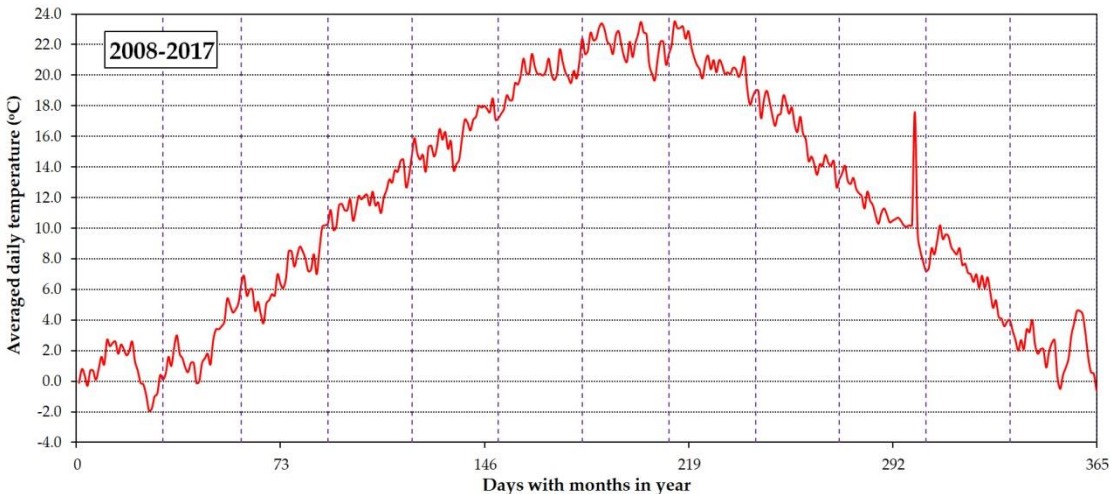

**Figure 5.** Average annual values of air temperature in the period from 2008 to 2017.

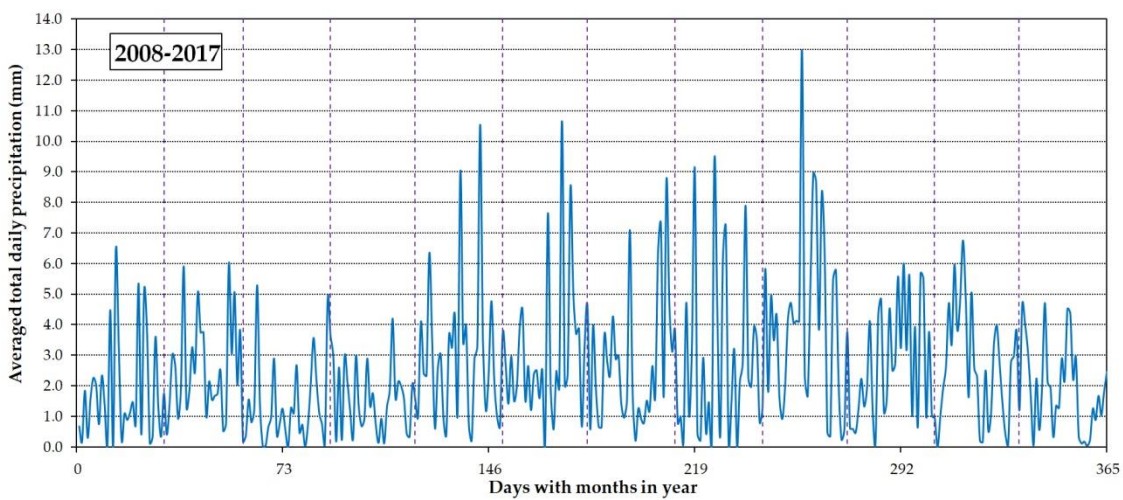

**Figure 6.** Averaged total daily precipitation in the period from 2008 to 2017.

## 5. Results and Discussion

Figure 7 shows the drought index $J_y$, calculated for a period of ten years.

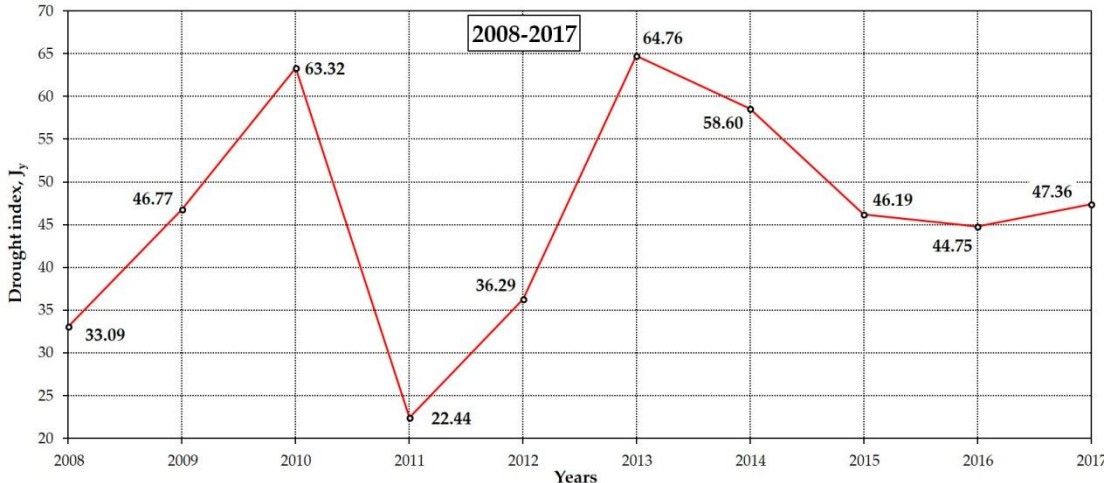

**Figure 7.** Index $J_y$ for ten observed years.

According to the values from Table 1, with the exception of 2011, which saw only a light drought, all other years had a normal degree of drought. It is obvious that the presented calculation has too big a time scale for defining of the period of clay excavation, because the normal or light drought degree for the entire year does not constitute useful information. Obtained results cannot be used for prediction of the period of clay excavation. Due to this, the next step would be the calculation of the drought index $J_m$ for each particular month of 10 observed years (Figures 8 and 9). The intention of this calculation is to obtain the detailed periods appropriate for clay excavation.

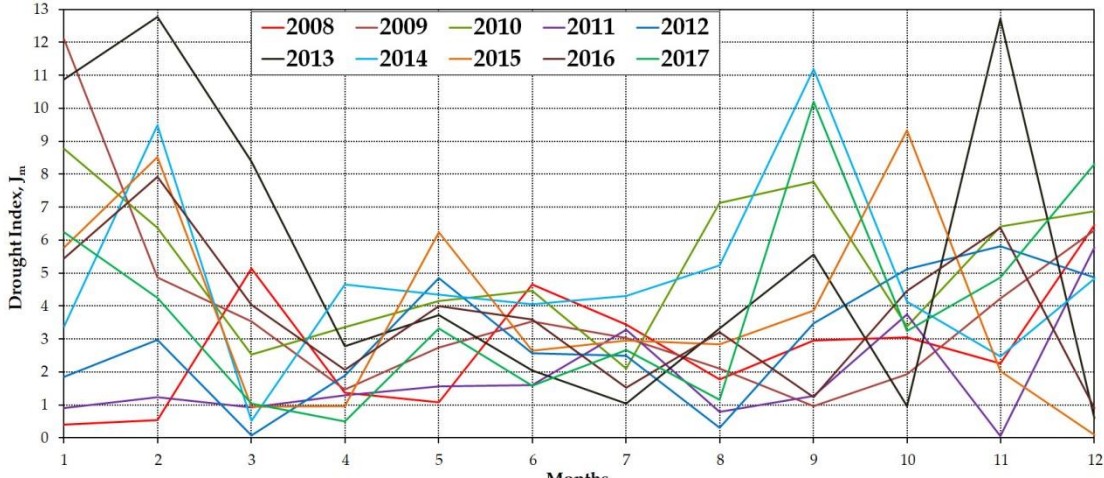

**Figure 8.** Drought index $I_m$ for each month of ten observed years.

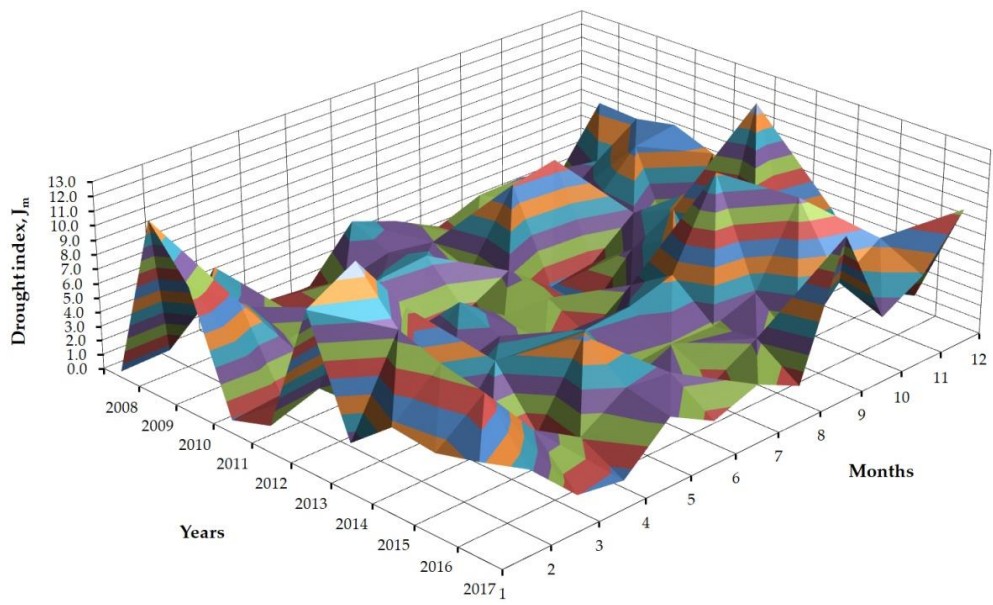

**Figure 9.** Three-dimensional presentation of the drought index $J_m$ for each month of ten observed years.

Although the values of the monthly drought index $J_m$ more precisely (compared with the yearly drought index $J_Y$) define the level of drought with respect of the associated period through the year, they cannot be used for defining of periods when clay excavation should be done. Changes of the drought level from moderate to extreme drought at the yearly level give a better insight into the variation of the drought period, but this also does not provide information about the optimum period of the clay excavation.

The same conclusion can be obtained with the monthly drought index $J_m$ for an averaged year, i.e., as an average from ten years from 2008 to 2017, Figure 10.

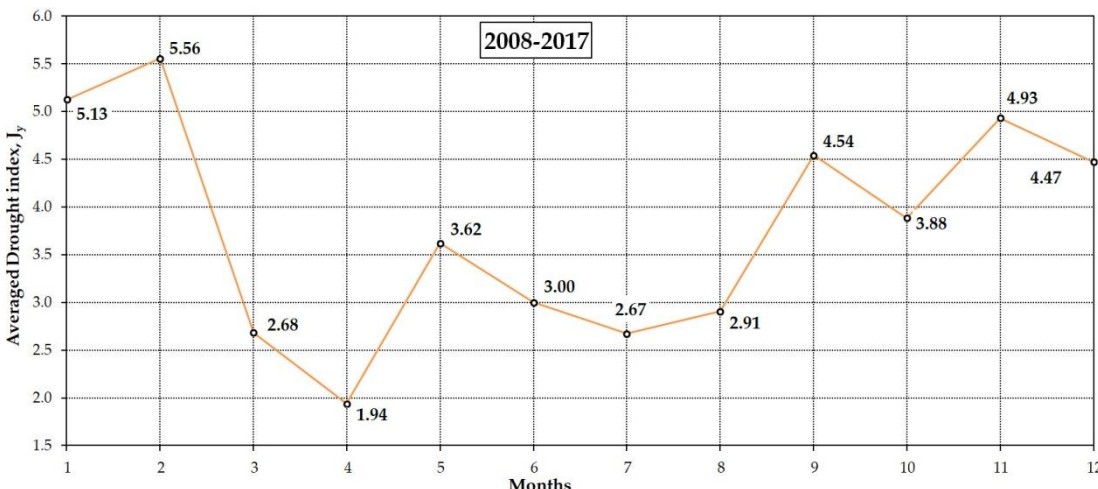

**Figure 10.** Monthly drought index $J_m$ for an averaged year.

Unfortunately, calculation of the yearly and monthly drought indexes does not give an insight into the detailed forecasting of the excavation. The RAPS analysis, based on the averaged data in Figure 5, using Equation (1), has defined three distinct periods throughout the year, divided from the beginning of January (1 January) until the beginning of April (10 April), then the beginning of October (11 October) until the end of the year. Figure 11 shows RAPS values for the average daily air temperature in the period between 2008 and 2017. If the analysis is carried out on a continuous basis, taking the year as an orientation period, the summer climate period and spring–autumn–winter can be separated. This, in fact, is the local climatic characteristic for the observed area.

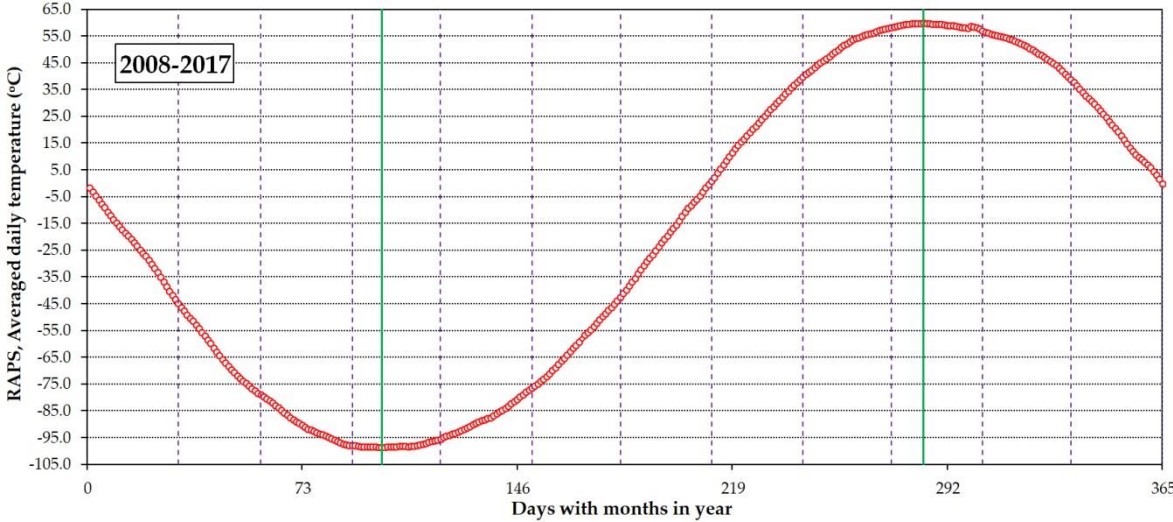

**Figure 11.** Rescaled adjusted partial sums (RAPS) for average daily air temperature in the period from 2008 to 2017.

Figure 5 shows the typical subset of average daily temperatures in the period between 2008 and 2017. However, taking into account the defined time periods of clay exploitation (from mid-April to mid-September), Figure 12 shows three subsets which coincide with the mentioned periods of time. The trends in the first two subsets are growing with a downfall seen in the last subset.

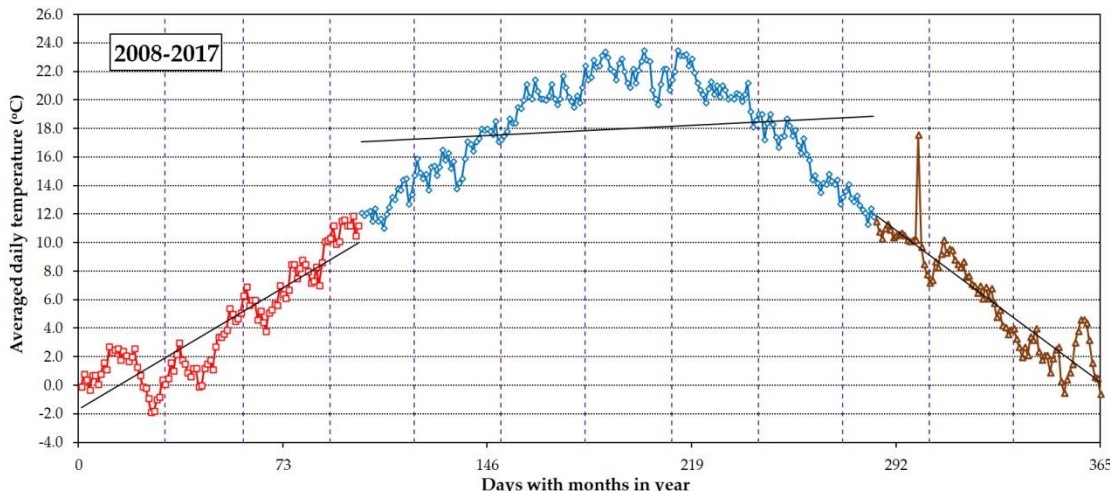

**Figure 12.** Typical subset of average daily air temperature in the period from 2008 to 2017.

Figure 12 shows climate characteristics, typical for the observed period, with regard to the average daily temperatures, as a dominant climate indicator. This is characterized by a secluded dominant period from spring until autumn, while the rest of the year shows an early spring period with the trend of the temperature increasing, as well as with the trend of temperature decreasing during the winter period.

Figure 13 shows RAPS values for the average total daily precipitation in the period from 2008 to 2017. Compared to the air temperature, the conducted RAPS analysis has identified several characteristic subsets.

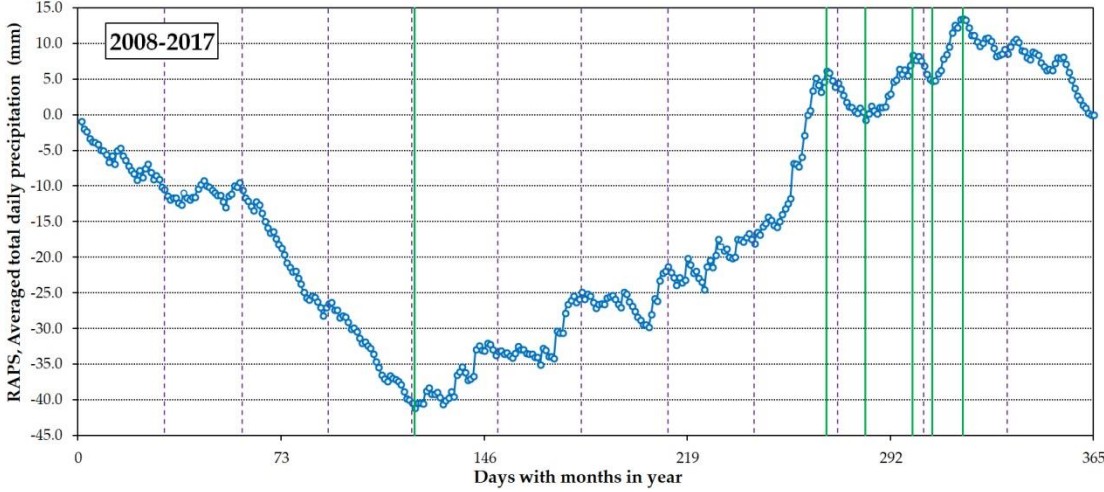

**Figure 13.** RAPS for the average total daily precipitation in the period from 2008 to 2017.

Figure 13 shows the typical subset of the average total daily precipitation for the period 2008–2017. Related to the air temperature and precipitation, the apparent elapsed overlap of the first and second distributed (divided) time periods becomes obvious. This is in accordance with the foreseen or defined prolongation of clay soil exploitation from mid-April to mid-September. Figure 14 shows that the four subsets represent the local climatic characteristics, i.e., the phenomenon associated with precipitation periods. At the air temperature, such short subsets have not been observed.

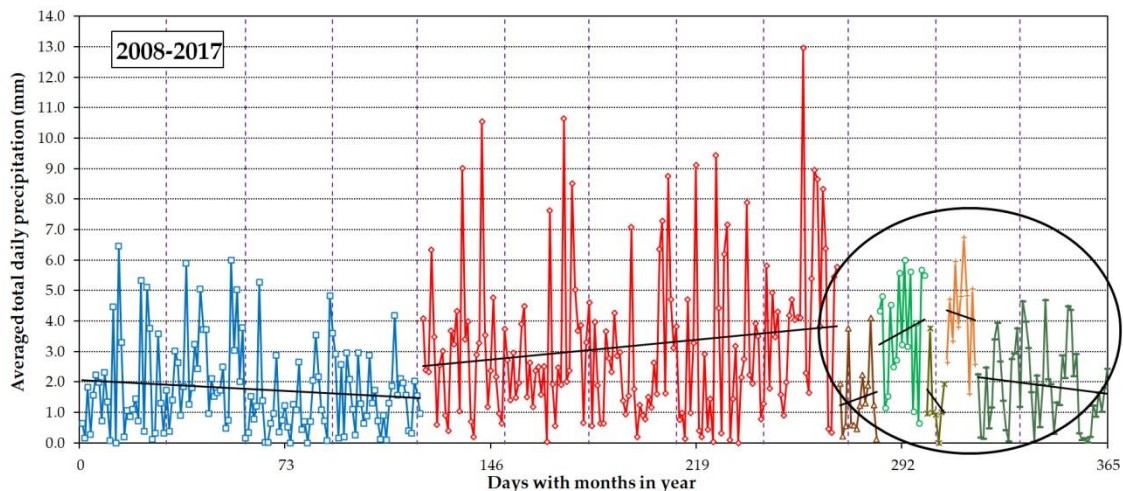

**Figure 14.** Typical subset of the average total daily precipitation in the period from 2008 to 2017.

However, between the end of September and mid-November, there have been four short interim subsets. Despite the defined period of exploitation of the clay soil, due to technical and technological reasons, clay exploitation continued between mid-September and the beginning of November.

It should be noted that maximum or minimum average air temperature, as well as total precipitation, is not prescribed by ordinance or legislation. Usually, the technical manager of the clay field defines the period for the excavation with regard of the current state of weather conditions, as well as weather forecasts. Of course, excavation would not begin in the case of the air temperatures under or close to zero or freezing conditions. Likewise, when the precipitation amount is such that clay field is soggy and unsuitable for the movement of the digging machines on the clay, excavation would not begin. Definition of the trends of the obtained subseries of total precipitation will be of benefit for planning and organization activities regarding clay excavation.

Figure 15 shows the isolated typical subsets of the average total daily precipitation in the period from 2008 to 2017, with four subsets also presented. In each case, clay soil exploitation is recommended during the first (26 September–10 October) and, especially, during the third (27 October–3 November) period.

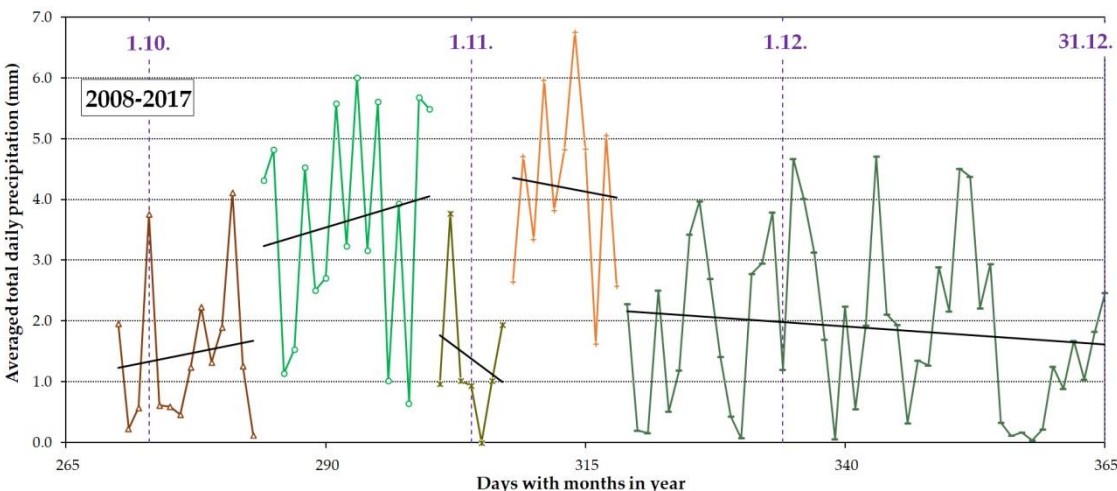

**Figure 15.** Featured typical subset of the average total daily precipitation in the period from 2008 to 2017.

Given the lower average air temperature, in the period (subset) from 26 September (14.3 °C) to 03 November (8.3 °C), (Figure 15), the previous claim is highlighted, especially if the conditions of

the clay soil exploitation, i.e., the total daily precipitation are taken into account. In addition, values of temperature and precipitation allow clay excavation. The fifth period (from Figure 15), due to average air temperatures close to 0 °C, is the period of time which is definitely not appropriate for clay excavation, despite a decreasing trend of the total daily precipitation.

## 6. Conclusions

The analysis carried out on real case studies has shown that normal trend analysis is not relevant or applicable to defining clay exploitation periods. Further, predicting the clay exploitation period in view of the observed extreme values of the original time subsets of average daily air temperature and total daily air volume would also be unacceptable, because these values are not within typical interpenetration, which would give the wrong information to technologists and engineers working in situ.

The RAPS method has proven to be elegant and applicable to this kind of analysis, and the process is user-friendly. Such data clustering, which points to periodicities in average daily air temperatures and total daily precipitations, could not be seen by a simple analysis of the trend of observed data. The drought index $J$, despite his wide applicability and usage, in this particular but very common problem, did not fully prove to be justified. The presented model for determining the optimal period of clay exploitation is applicable to each location in the world, taking into account the climatic characteristics described over the time subsets of average daily air temperature and the total precipitation levels.

It could be seen that clay excavation could be planned in accordance with the usual engineering practice knowledge and experience, which is (generally) from April to October (including October) for Croatia. Such a statement is supported by the presented model, i.e., methodology. It should be noted that the presented method separates time period(s) where periods of drastic climate change are heavily emphasized, which in this case is the period from the beginning of October to the middle of November. This is very important because the aforementioned time period is usually the final stage of the excavation period during the year, i.e., ending the technological process of the clay excavation and defining the financial balance for the entire year. In that way, managers and field engineers have information when they can expect weather changes, in this case precipitation deviations, which can disable excavation.

The reliability of the described model depends on climate change, which, unfortunately, becomes increasingly negative where extreme temperatures, precipitation, and other meteorological indicators are difficult to foresee. Even complex forecasting time models or weather forecasts are not reliable, but when considering the clay exploitation period or the start of the excavation in clay fields, the local weather forecasts should be considered.

**Author Contributions:** Conceptualization, D.T.; Data curation, B.S.; Formal analysis, B.Đ.; Investigation, D.T.; Methodology, B.Đ.; Project administration, N.K.; Resources, D.T.; Software, B.Đ.; Supervision, B.S.; Validation, B.S.; Visualization, B.Đ. and N.K.; Writing—original draft, D.T.; Writing—review and editing, N.K.

**Funding:** The APC was funded by Office of Certified Civil Engineer Božo Soldo, Vinka Mederala 4b, 42000 Varaždin, Croatia.

**Acknowledgments:** Publication process is supported by the Office of Certified Civil Engineer Božo Soldo, Vinka Mederala 4b, Varaždin 42000, Croatia.

**Conflicts of Interest:** The authors declare no conflict of interest.

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
