# Peer review of "Impact of Seasonal Changes of Precipitation and Air Temperature on Clay Excavation"

_sustainability, doi:10.3390/su11226368_

Round 1

Reviewer 1 Report

The authors analyzed the measurements of precipitation and temperature from the selected area. However, after reading the whole article, I am not convinced whether they managed to achieve the goal assumed at the beginning, i.e. the optimization of the clay exploitation period. Authors should also improve the style of the article. In the text there are repeated many times very short 1-2 sentence paragraphs. The following paragraphs are not related to each other and form some kind of collage. Specific comments are set out below:

1) Line 29 - What do the authors support the statement that clay excavation is the most important economic branch in the world?

2) The introduction needs to be rearranged. The following paragraphs are loosely interlinked and do not form a coherent whole.

3) Line 196 - The authors first write about the method of exploitation and in the next paragraph write about the results of measurements. How does this relate to the previous paragraph?

4) Figures 3 and 5 have the same description although they represent a different data.

5)  Lines 208, and 210, and 2013 - repetition of what is shown in the figures and unnecessary new paragraph.

6) Some diagrams on the horizontal axis show days and other months - this makes it very difficult to compare the data.

7) Line 240 – why from April till August, not March till August? On what grounds was it concluded that this was so?

8) Figure 12 - The fitting of trend lines in the beginning and end of the results does not raise any objections. However, the middle line deviates dramatically from the data. In the discussed range the average temperature varies from 12 to 24 degrees and this is described by a line of about 18 degrees. In addition, during this period, the temperature diagram resembles a parabola and is described in a straight line, which can’t be correct. Please comment.

9) There are no clear conclusions from the analysis presented. In what time can clay excavation in the area be planned?

Author Response

Response to Reviewer 1 Comments

Point 1: Line 29 - What do the authors support the statement that clay excavation is the most important economic branch in the world?

Response 1: Statement is reformulated in the text.

Point 2: The introduction needs to be rearranged. The following paragraphs are loosely interlinked and do not form a coherent whole.

Response 2: The introduction has been corrected and the sentences now form a better whole.

Point 3: Line 196 - The authors first write about the method of exploitation and in the next paragraph write about the results of measurements. How does this relate to the previous paragraph?

Response 3: Line 195 is the new subtitle. Line 196 shows the inputs for further calculation of the observed area (Average daily air temperatures in the period from 2008 to 2017 and Total daily precipitation in the period from 2008 to 2017).

Point 4: Figures 3 and 5 have the same description although they represent a different data.

Response 4: Renamed description of the figure in the text.

Point 5: Lines 208, and 210, and 2013 - repetition of what is shown in the figures and unnecessary new paragraph.

Response 5: Corrected in text.

Point 6: Some diagrams on the horizontal axis show days and other months - this makes it very difficult to compare the data.

Response 6: All images have been modified in the text.

Point 7: Line 240 – why from April till August, not March till August? On what grounds was it concluded that this was so?

Response 7: The claim in mentioned sentence is unfounded.  Sentence from line 239 – 241 will be erased.

Point 8: Figure 12 - The fitting of trend lines in the beginning and end of the results does not raise any objections. However, the middle line deviates dramatically from the data. In the discussed range the average temperature varies from 12 to 24 degrees and this is described by a line of about 18 degrees. In addition, during this period, the temperature diagram resembles a parabola and is described in a straight line, which can’t be correct. Please comment.

Response 8: Defining of the linear trend is common procedure within RAPS method, which is explained by lines 132-134. But, due to the comments of the reviewer, additional part will be added after the ''subseries'' at the end of sentence on line 134: ''which is usual procedure within RAPS method'' for better understanding.

Point 9: There are no clear conclusions from the analysis presented. In what time can clay excavation in the area be planned?

Response 9: It could be seen that clay excavation could be planned accordingly to the usual engineering practice knowledge and experience, which is (generally) from April to October (including October) for Croatia. Such statement is supported by presented model, i.e. methodology. It should be noted that presented method separate time period(s) where periods of drastic climate change are heavily emphasized, which is this case period from beginning of October till the middle of November. This is very important because mentioned time period is usually the final stage of the excavation period during the year, i.e. ending the technological process of the clay excavation and defining the financial balance for the entire year. In that way, managers and field engineers have information when they can expect weather changes, in this case precipitation deviations, which can disable excavation.

Mentioned statements would be added as a new paragraph after the second paragraph.

Reviewer 2 Report

This is a very interesting paper. The concept and motivation are clear and the paper follows a logical order. I like the idea of applying hydrological concepts to what is usually thought of as a geotechnical or mining problem. 

I have written extensively on the pdf manuscript, please read through the changes and comments as there are many (a total of 203). I think the term "quarry" or clay quarry is better than clay soil exploitation area. I used the term clay deposits as well. plural of quarry is quarries

Readers would benefit from a more clear introduction. You state that the "excavation season" runs between mid-April and mid-September because the clay is difficult to excavate in late Autumn, Winter, early Spring, but you do not say precisely why. Is it always too wet during the off season, or too dry? Or both? Obviously it may be too cold as well, but you only mention that at the end of your conclusions. So, clearly state the specific condtions you are trying to avoid and the conditions you would find acceptable. 

I understood your calculations until you began doing averages. There are many ways to take averages for time series data, especially temperature and rainfall. Be very clear how you compute the monthly, yearly averages for all of your data. By the way, you don't say how you got so much weather data. Is it recorded at the quarry? Is there a weather station nearby (how near)?

Some of the graphs need to be re-drawn. 

Figure 7 needs larger text. 

Figure 8 needs larger text for axis labels and better line quality of the graphs so that the reader knows which line represents 2014 or 2016 or 2017...they all look grey, even at 200% zoom. 

Figure 9 should be rotated on the Z-axis so the reader's point of view is more looking from the "Years" axis down the centerline of the "mountain range" in order to see all the data. Too much is hidden. 

Figure 10. Why do you have numbers with each data point? The numbers are not very precise compared to the graph, so why put them on ? Example, the second data point is plotted at about 5.53 but is labeled with a 6. 

Figure 12. You never really stated how you determined if a subsequence could have a trend line. Was there some quantitative criteria or was it visual based on the graph?

Figure 13. How did you perform a RAPS calculation on average total daily precipitation? Did you first average the daily precipitation then RAPS? How did you account for the variability of the rainfall for a given day? You have 10 values for each day and those values would have a deviation that would not be the same. Just state how you performed the calculation on averaged data. 

Author Response

Response to Reviewer 2 Comments

Point 1: I have written extensively on the pdf manuscript, please read through the changes and comments as there are many (a total of 203). I think the term "quarry" or clay quarry is better than clay soil exploitation area. I used the term clay deposits as well. plural of quarry is quarries.

Response 1: The name of the "quarry" is primarily used for open pits where the technical and construction stone (amphibolite, andesite, basalt, diabase, granite, dolomite, limestone), architectural stone and mineral raw materials (Iron and Ferro-Alloy Metals, Non-Ferrous Metals, Precious Metals, Industrial Minerals, Mineral Fuels) are exploited. And the name "exploitation field" is used for open pits where clay, construction sand and gravel is exploited.

Point 2: I understood your calculations until you began doing averages. There are many ways to take averages for time series data, especially temperature and rainfall. Be very clear how you compute the monthly, yearly averages for all of your data. By the way, you don't say how you got so much weather data. Is it recorded at the quarry? Is there a weather station nearby (how near)?

Response 2: Weather station is located about 4.3 km northeast of the exploitation field.

Point 3: Figure 7 needs larger text..

Response 3: Image have been modified in the text.

Point 4: Figure 8 needs larger text for axis labels and better line quality of the graphs so that the reader knows which line represents 2014 or 2016 or 2017...they all look grey, even at 200% zoom.

Response 4: Image have been modified in the text.

Point 5: Figure 9 should be rotated on the Z-axis so the reader's point of view is more looking from the "Years" axis down the centerline of the "mountain range" in order to see all the data. Too much is hidden.

Response 5: Image have been modified in the text.

Point 6: Figure 10. Why do you have numbers with each data point? The numbers are not very precise compared to the graph, so why put them on ? Example, the second data point is plotted at about 5.53 but is labeled with a 6.

Response 6: Image have been modified in the text.

Point 7: Figure 12. You never really stated how you determined if a subsequence could have a trend line. Was there some quantitative criteria or was it visual based on the graph?

Response 7: Defining of the trend line is a usual procedure within RAPS, which is stated in text from line 132. This is done to simplify the presentation of growth or decrease in values of the subseries.

Point 8: Figure 13. How did you perform a RAPS calculation on average total daily precipitation? Did you first average the daily precipitation then RAPS? How did you account for the variability of the rainfall for a given day? You have 10 values for each day and those values would have a deviation that would not be the same. Just state how you performed the calculation on averaged data.

Response 8: First step was averaging of the precipitation. The reviewer is correct in claiming that there are daily fluctuations in precipitation values. Available precipitation data were only total daily precipitations. Also, hourly analysis definitely would not have justification, because such analysis does not have a purpose, due to the large number of data for analysis, as well as for the final results.

Round 2

Reviewer 1 Report

Thank you for all comments. The corrections significantly improved the quality of the article.